# Lyme Borreliosis Serology: A Prospective Cohort Study of Forestry Service Workers in the Netherlands over 8 Years (2008 to 2016) of Follow-Up

**DOI:** 10.3390/life13051143

**Published:** 2023-05-08

**Authors:** Hadewych ter Hofstede, Jeroen Haex, Michael Belias, Marije Oosting, Leo A. B. Joosten, Foekje F. Stelma

**Affiliations:** 1Department of Internal Medicine, Radboud Center for Infectious Diseases (RCI), Radboud University Medical Center, 6500 HB Nijmegen, The Netherlandsleo.joosten@radboudumc.nl (L.A.B.J.); 2Department of Medical Microbiology, Radboud University Medical Center, 6500 HB Nijmegen, The Netherlands; j.haex@dji.minjus.nl (J.H.); foekje.stelma@radboudumc.nl (F.F.S.); 3Department for Health Evidence, Radboud Institute for Health Sciences (RIHS), Radboud University Medical Center Nijmegen, Mailbox 133, P.O. Box 9101, 6500 HB Nijmegen, The Netherlands

**Keywords:** Lyme borreliosis, *Borrelia burgdorferi*, tick exposure, serology, prevalence, forestry workers

## Abstract

There is little known about the dynamics within responses to *Borrelia* spp. upon repeated exposure to tick bites and the development of serological markers over time. Most studies have investigated antibody development in risk populations over a short period of time. Therefore, we aimed to study the dynamics of anti-*Borrelia* antibodies in forestry service workers over 8 years in association with tick bite exposure. Methods: Blood samples from 106 forestry service workers originally included in the 200 Functional Genomics Project (Radboudumc, Nijmegen, the Netherlands) were followed for 8 years and tested annually for anti-*Borrelia* antibodies (ELISA and Western blot). IgG seroconversion was related to the number of tick bites in the previous year, which was obtained through annual questionnaires. The hazard ratio for *Borrelia* IgG seroconversion was calculated using Cox regression survival analysis and a logistic regression model, both adjusting for age, gender and smoking. Results: Borrelia IgG seropositivity in the study population did not vary significantly between years and the average prevalence was 13.4%. Of the 27 subjects that underwent seroconversion during the study period, 22 reconverted from positive to negative. Eleven subjects seroconverted a second time. The total seroconversion rate per year (negative to positive) was 4.5%. Active smoking was associated with IgG seroconversion in the >5 tick bites group (*p* < 0.05). According to the two models used, the risks of IgG seroconversion in the >5 tick bites group were HR = 2.93 (*p* = 0.10) and OR = 3.36 (*p* < 0.0005). Conclusions: *Borrelia* IgG seroconversion in forestry service workers was significantly related to increasing tick bite exposure in a survival and logistic regression model adjusting for age, gender and smoking.

## 1. Introduction

Lyme borreliosis is an infectious disease that is caused by spirochetal bacteria of the genus *Borrelia*. The bacterium is transmitted to humans by the bite of infected ticks of the genus *Ixodes*. Approximately 20% of all ticks in the Netherlands are infected with *Borrelia* spp. [1]. *Borrelia afzelii* and *Borrelia garinii* are the most common species found in the Netherlands. A tick bite causes active Lyme borreliosis in about 0.3–5.2% of exposed cases [2]. In the Netherlands, the incidence of erythema migrans increased from 39/100,000 inhabitants in 1994 to 134/100,000 inhabitants in 2009 [3]. Since 2009, the incidence has remained the same at approximately 140 cases per 100,000 inhabitants [4]. *Borrelia* diagnosis relies upon serology (two-tier protocol with ELISA and Western blot), which attempts to determine recent or past exposure to *Borrelia* [5]. Unfortunately, it is not possible to distinguish this exposure accurately. Still, serology is the most reliable and validated diagnostic method currently used for Lyme borreliosis. Seroconversion was defined as a change in IgG status from negative to positive in two consecutive years. Not much is known about the dynamics of humoral and cellular immune responses to *Borrelia* spp. upon repeated exposure to tick bites [6]. Additionally, little is known concerning the extent of complaints and development of serological markers over time. The main aim of this prospective study was to describe the relationship between exposure to tick bites over 8 years and the development of *Borrelia*-specific IgG responses in a population of forestry workers who were repeatedly exposed to tick bites. The relationship between tick bite exposure and cellular immune responses against *Borrelia* spp. was also studied.

## 2. Materials and Methods

### 2.1. Subjects

This trial was approved by the Medical Ethics Review Committee, CMO Arnhem-Nijmegen (Registration Number 2011/399). A total of 312 subjects was included in the 200 Functional Genomics Project (www.humanfunctionalgenomics.org). A subsample of 106 individuals (88 men and 18 women), aged 44 ± 11 years (mean ± SD), was included in this study (Figure 1). The inclusion of participants took place from 2008 to 2009 and involved follow-up for 8 years. Unfortunately, the follow-up in 2012 was not performed due to logistical issues. All participants worked in regions that were highly endemic of ticks (forests and national parks in the central part of the Netherlands) (Figure 2). Participants did not have the same forestry exposure; although most worked outside as forestry workers, some had more administrative duties. All forestry workers obtained tick-repellent clothing from their employer, which was not separately analyzed.

Only subjects that took part in at least 7 out of the 8 rounds of annual blood sampling and had filled in the annual questionnaires were included. Participants with chronic co-morbidities were excluded in order to minimize interfering bias with the immune responses. Information was gathered by the questionnaires concerning age, gender, smoking, the individuals’ average amount of tick bites per year, their average amount of forest exposure per day (3 groups: <1.5 h/day, 1.5–5 h/day and >5 h/day outside work) and the presence of Lyme borreliosis-associated symptoms (LAS). Venous blood was collected from all subjects in EDTA and serum tubes (BD Vacutainer, N107417-1880; BD Life Sciences, Becton Drive, Franklin Lakes, NJ, USA) for antibody detection and measurement of cellular immune functions.

### 2.2. Lyme Borreliosis-Associated Symptoms (LAS)

Participants completed a yearly questionnaire concerning any symptoms they had experienced during the past year that could be related to Lyme borreliosis. The symptoms included memory loss; headaches; fatigue; myalgia and arthralgia; cardiac arrhythmia; neurological symptoms; and discoloration or size differences in the feet, legs or ears.

### 2.3. Antibody Detection

Testing for antibodies in *Borrelia* spp. was conducted by a two-tier protocol using an enzyme-linked immunosorbent assay (ELISA, SERION ELISA classic; Virion/Serion, Würzburg, Germany) and a Western blot (EUROLINE-WB; Euroimmun, Lübeck, Germany) according to the manufacturer’s instructions. The ELISA used native antigens based on a mixture from the European strain of *Borrelia afzelii* and *B. garinii* and recombinant VlsE. Results were coded as positive, negative or equivocal for IgM and IgG. Semi-quantitative data for IgG antibody levels were also available. Western blot was used to test for IgM and IgG reactivity against a whole cell antigen extract (*B. afzelii* and *B. garinii*) and recombinant VlsE as an early marker across all *Borrelia* species. IgM reactivity against specific antigens (native p83, p39, p31, p30, p25, p21, p19, p17 and recombinant VlsE) was interpreted to be positive if p25 was positive, and when in doubt, at least one other specific band was positive. IgG was interpreted as positive if VlsE was positive, and when in doubt, at least two other specific bands were positive. Results were coded as positive, negative or equivocal for IgM and IgG. In this research setting, both ELISA and Western blot assays were performed on every participant sample and also if the initial ELISA was found negative. Participants with positive Western blots were classified as antibody-positive.

In this study, we did not look for cross-reactivity with other infections; however, it is known that *Borrelia* serology cross-reacts with the serology of tick-born relapsing fever, Rocky Mountain spotted fever (RMSF), syphilis, *Anaplasma* spp. and *Ehrlichia* spp. In daily clinical practice, these cross-reacting infections are all very uncommon, except for syphilis.

*Borrelia* IgG seroconversion was defined as a change in IgG status from negative to positive between two consecutive years taking in account the combined results from the IgG ELISA and Western blot assays.

### 2.4. Cellular Immune Functions

The isolation of PBMCs was performed using Ficoll isolation as described by Oosting et al. [7]. Tests on an RPMI culture medium were added as a negative control. Data showing a high response in the RPMI group, which was an indicator of contamination, were excluded. Testing for cytokine responses was performed using commercial ELISA kits (PeliKine Compact, Sanquin, Amsterdam; R&D Systems, Minneapolis, MN, USA) according to the manufacturer’s instructions. Data on immunological responses were available for the following cytokines: IL-1β, IL-6, IL-10, IL-17, IL-22, TNF-α and IFN-γ. When looking at the IL-1β group, 24 h PBMC data were available for all 8 years. For the IL-6 and TNF-α groups, 24 h PBMC data were available for 6 years. In the IL-10 group, 24 h PBMC data were available for 5 years. For the IL-17 group, 7-day PBMC data were available for 3 years. Additionally, for the IL-22 and IFN-γ groups, 7-day PBMC data were available for 2 years.

### 2.5. Statistical Analysis

Proportions were compared by chi-squared analysis. The time to IgG seroconversion was studied by Kaplan–Meier survival analysis.

Further, a Cox proportional hazards model was fitted; since we had repeated measurements, we inputted the missing values in our dataset using either the last observation carried forward or the next observation carried backward technique. Particularly, if the missing value occurred during years 2–8, we used the first technique, while if the missing value was observed in the first year, the second technique was used.

Finally, to control for consistency, a logistic regression model, adjusted for age, gender and smoking, was performed.

## 3. Results

Population characteristics, as well as the general serology outcome of IgG and IgM, the number of tick bites and forest exposure for each year of follow-up, are presented in Table 1. Unfortunately, no data for 2012 were obtained. Furthermore, data on forest exposure from 2015 were also not obtained. A total of 106 subjects (median age 46, range 21–65) was included in this study. Eighty-eight (83.0%) subjects were male and eighteen (17.0%) subjects were female. On average, over all the years, 29.1% had zero tick bites, 28.8% had 1–5 tick bites and 28.9% had >5 tick bites, with 13.2% missing data. In the >5 tick bites group, the percentages varied between 17.9% (2013) and 34.9% (2016) (see also Figure 3). IgG seropositivity remained stable at approximately 13.4%, while IgM positivity showed more year-to-year variation. On average, 42.8% of the participants reported Lyme borreliosis-associated symptoms (LAS).

### 3.1. IgG Seropositivity in Relation to Tick Bites

Subjects exposed to more than five tick bites were more often IgG-seropositive when compared to subjects not exposed to tick bites (Figure 3). This difference was especially significant during 2008, 2014, 2015 and 2016 (*p* < 0.05).

### 3.2. Characteristics in Population That Seroconverted for IgG Compared to Population That Did Not Seroconvert

Of the 106 subjects, 25% (n = 27) showed seroconversion from negative to positive at least once. The remaining 79 subjects were either seronegative (71%; n = 75) or seropositive (4%; n = 4) during the 8 years. Most seroconverting participants (59%) experienced more than five tick bites during the previous year of their first seroconversion. Of the 27 subjects that underwent seroconversion, 22 reconverted from positive to negative. A second seroconversion took place in 11 subjects. In the year before seroconversion, they were exposed to tick bites for an average of more than five times. The mean seroconversion rate per year (negative to positive) remained around 4–5%.

No significant differences were found between the baseline characteristics of the participants that seroconverted and those that did not, as seen in Table 2. In the >5 tick bites group, significantly more subjects were seroconverted compared to those that did not seroconvert.

### 3.3. Risk of Seroconversion within Tick Bite Groups

The risk of seroconversion in the zero tick bites group was estimated to be 0.085 on average. Relative to this category, the 1–5 tick bites group showed on average a 1.23% lower risk of seroconversion (non-statistically significant result), while the >5 tick bites group showed 171.91% more risk of seroconversion compared to the zero tick bites group (*p* < 0.001).

### 3.4. Stratified Analysis—The Influence of Age, Gender and Smoking on IgG Seroconversion

The risk of seroconversion in the zero tick bites group was lower in females than in males (83% less), while in the 1–5 and >5 tick bites groups, the risk increased in females between 5.6 and 7 times on average, respectively. Similarly, smokers showed a lower risk of seroconversion compared to non-smokers in the zero tick bites group (32% less), while in the 1–5 and >5 tick bites group, the risk of seroconversion increased in smokers by three times. However, overall, the correlation between seroconversion and smoking was not significant (r = 0.03; *p* = 0.78). Finally, for age, the risk of seroconversion in the various tick bite groups did not show a clear trend. It appeared that in the zero tick bites group, the risk of seroconversion increased with age, but if exposure to tick bites occurred (1–5 and >5 tick bite groups), the risk of seroconversion decreased with age.

### 3.5. Survival Analysis

The mean time to IgG seroconversion in the group exposed to zero tick bites (reference) was 6.80 years (95% CI; 5.72–7.88 years); in the group exposed to 1 to 5 tick bites, it was 6.48 years (95% CI; 5.66–7.30 years); and in the group exposed to >5 tick bites, a mean time of 5.12 years (95% CI; 3.98–6.26) emerged. The survival distribution for the three tick bite subgroups was statistically and significantly different, of χ^2^ = 8.754 and *p* < 0.05 (Mantel–Cox; Figure 4).

The full multivariate Cox regression model, adjusted for age, gender and smoking as well as taking time in consideration, yielded the following hazard ratios for the different tick bite exposure groups: HR = 1.68 (95% CI; 0.49–5.73; *p* = 0.41) when comparing the 1–5 tick bites group to the reference group (zero tick bites) and HR = 2.93 (95% CI; 0.81–10.58; *p* = 0.10) when comparing the >5 tick bites group to the reference group.

### 3.6. Logistical Regression Model Investigating IgG Seroconversion in Relation to Tick Bites

A logistic regression analysis adjusted for age, gender and smoking was performed to validate the results obtained by the survival analysis. In this analysis, each year was treated as an individual event. IgG seroconversion was significantly associated with the >5 tick bites group (X^2^ = 35.6; *p* < 0.0005). IgG seroconversion was not associated with the 1–5 tick bites group (X^2^ = 1.06; *p* = 0.87). Being exposed to more than five tick bites per year was significantly associated with an increased risk for IgG seroconversion (OR = 3.36; *p* < 0.0005).

### 3.7. Cellular Immune Response and Tick Bites

No statistically significant variation in cellular immune functions measured by the PBMC production of IL-1β, IL-6, IL-10, IL-17, IL-22, TNF-α and IFN-γ was observed between the three tick bite subgroups (Figure 5). When comparing these immunological parameters to IgG response, no association was observed.

## 4. Discussion

This study is the first to show a positive association between the number of tick bites per year and IgG seropositivity for *Borrelia* species over several years. In particular, high numbers of tick bites (>5 per year) were associated with an increased rate of IgG seroconversion. Further, age, gender and smoking were identified as possible effect modifiers. The number of tick bites per year did not influence cellular immune response. We chose to categorize the number of tick bites into three categories (0, 1–5 and >5 tick bites). However, when further subdividing the number of tick bites applied from 6 to 12 and more than 12 tick bites per year, we found a further increased risk of seroconversion in the group of 6 to 12 tick bites per year compared to the one with no tick bites. As mentioned earlier, the >5 tick bites group demonstrated a significantly increased risk of seroconversion compared to the reference. This appeared to correspond with the fact that approximately one in five ticks is infected by *Borrelia* spp. in the Netherlands [1].

IgG seropositivity in Dutch forestry workers (13.4%) was higher than IgG seropositivity in the general population (6%) but lower than expected when considering the prevalence of infected ticks (20%). This is in contrast to research performed in Poland; there, an IgG seropositivity of 20% was observed in a highly exposed population of forestry workers [8,9], which is much higher than the population IgG seropositivity of below 5% in this study [10]. Only around 6–15% of ticks in Poland are infected with *Borrelia* spp., which would result in a lower risk due to tick bite exposure than in the Netherlands [11]. A recently published Belgian study also showed a higher seroprevalence (21.6%) among forestry workers [12]. In Belgium, around 12% of all ticks are infected with predominantly *B*. *afzelii* and *B*. *garinii* [13]. The low IgG seroprevalence in Dutch forestry workers may be explained by a higher awareness of checking for tick bites in combination with the implementation of protective clothing. The type of forest may also influence infection rates. Recent research by Kiewra et al. (2018) showed that deciduous and mixed-deciduous forests have a significant impact on IgG seropositivity to *Borrelia* spp. [9].

The seroconversion rate in our population was low, at around 4–5%, with sporadic peaks at around 9%. This is in accordance with the research previously performed by Fahrer et al. (1988), where a seroconversion rate of 7% was observed among Swiss orienteers (‘orienteering’ refers to a competitive cross-country sport with a map and a compass, usually taking place in forests), frequently exposed to ticks [14]. We know from our population that people with high exposure to ticks and high IgG *Borrelia* positivity worked in a forest with a substantial amount of game and therefore animals potentially carrying ticks (i.e., deer, wild pigs).

IgM reactivity proved to be a poor marker due to significant year-to-year fluctuations. Others supported the observation that IgM is a poor marker for *Borrelia* exposure [15]. A total of four subjects presented with persistent positive IgG results throughout the whole study period (persistent positive ELISA and Western blot results). About 20 subjects showed periods of generally high IgG titers in the IgG ELISA (around 40–100 U/mL), whereas the cut-off point for a positive result was >5 U/mL. It was difficult to determine if this took the form of active disease or if this may have suggested some form of memory response upon repeated exposure, as previously suggested by Glatz et al. [16]. Further, many of the observed seroconversions and reconversions were around the test cut-off point and could be explained by simple test fluctuations around the cut-off caused by low IgG titers. However, this could not explain seroconversion and reconversion entirely, as the ELISAs always had to be confirmed by Western blot positivity in order to be classified as positive. Therefore, we believe that seroconversions just above the cut-off were true seroconversions. Another argument is that seroconversions were associated with tick bite exposure. We considered a second seroconversion to be a reinfection. Other researchers were able to demonstrate that seroconversion was due to reinfection with another strain of *Borrelia* [17,18]. We were not able to distinguish between reinfection with various *Borrelia* subspecies as serology is highly cross-reactive between *Borrelia* subspecies. The normal clinical procedure is to perform a Western blot test after a positive ELISA result. In our study, Western blot was also performed in cases of negative ELISA screening tests. The combination of a negative ELISA with a positive Western blot was defined as a positive result for Lyme borreliosis. Remarkably, in this research, 13% of negative ELISA samples turned out to be positive in the Western blot assay with clear *Borrelia*-specific bands, such as VlsE. Our data showed a trend toward higher IgG seroconversion rates within the smoker group when compared to the non-smoker group. This is in accordance with the research previously conducted by Brandsma et al. and Calapaiet et al. [19,20]. A hypothesis from this research was that smoking activates the immune system due to exposure to oxidative stress, consequently leading to increased memory B-cell production. However, in our study, the sample size was insufficient to make clear statements about the role of smoking in relation to IgG seroconversion. An association was seen between IgG seroconversion and age, as the risk of seroconversion appeared to decline slightly with increasing age. This observation may be explained by the true decline in immune function with age or a decrease in forest exposure in older participants; however, this last argument was not observed in this research. Recently published Belgian research however has shown that seroprevalence is higher in forestry workers that visit the forest more frequently [12].

As mentioned before, the amount of tick bites did not influence cellular immune response. All tested cytokines showed increased production when exposed to live *Borrelia* in concurrence with the research previously conducted by Jansky et al. [21]. Against expectations, no association between positive IgG and heightened immune response was observed. From September 2013, the forestry services gradually started to implement tick-repelling clothing (Rovince, Enschede). This clothing was treated with the insecticide permethrin. Interestingly, forestry service workers noted almost no changes in the number of tick bites they received, and no significant decreases in their IgG seropositivity rates were observed after implementing the new clothing. Previous research suggests that permethrin-treated clothing is effective, but the effectiveness is very dependent on the type of clothing being treated [22]. The most effective are socks and sneakers treated with insecticide, and the least effective appear to be shorts and shirts. As seen in Figure 3, a certain number of cases that reported no tick bites was still seropositive for *Borrelia* exposure. These participants may have seroconverted during an earlier period before this study. Another explanation is that the tick bites went unnoticed and the tick detached before being noticed or that participants forgot to mention the tick bite on the questionnaire [23]. Finally, the serological results did not match the physical symptoms that people reported. Additionally, objectivated erythema migrans did not always result in seropositivity; serological response may not have occurred due to early antibiotic treatment.

## 5. Conclusions

A high-risk population of forestry service workers only showed a slightly higher seroprevalence of *Borrelia* spp. when compared to the Dutch general population but the seroprevalence appeared to be lower than expected when considering the prevalence of infected ticks. Seroconversion was significantly related to higher tick exposure. Our data show a trend toward a higher IgG seroconversion rate within the smoker group and an association between increasing age and a slight decline in the risk of seroconversion. Finally, no correlation was observed between the number of tick bites and cellular immune responses against *Borrelia* species.

## Figures and Tables

**Figure 1 life-13-01143-f001:**
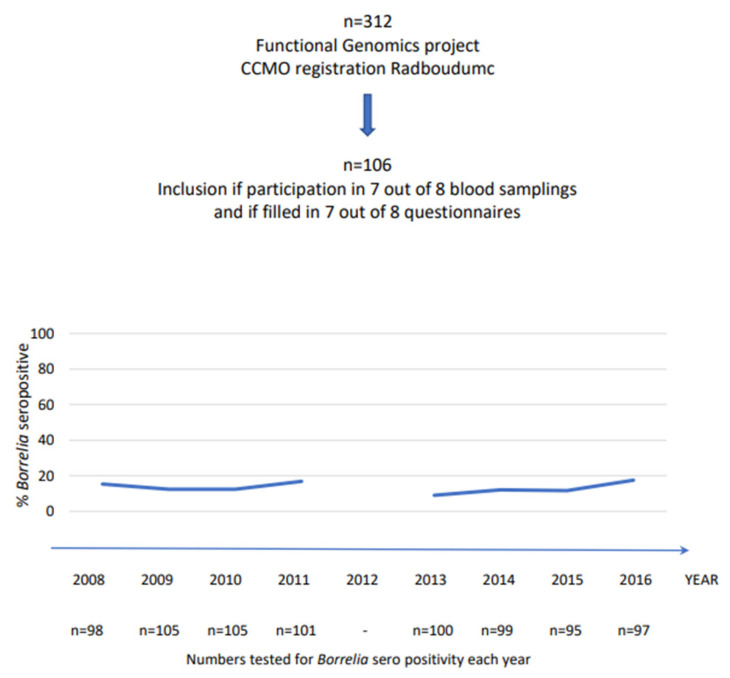
Schematic of the experimental design of the prospective cohort study of Borrelia IgG seroprevalence in forestry workers over 8 years.

**Figure 2 life-13-01143-f002:**
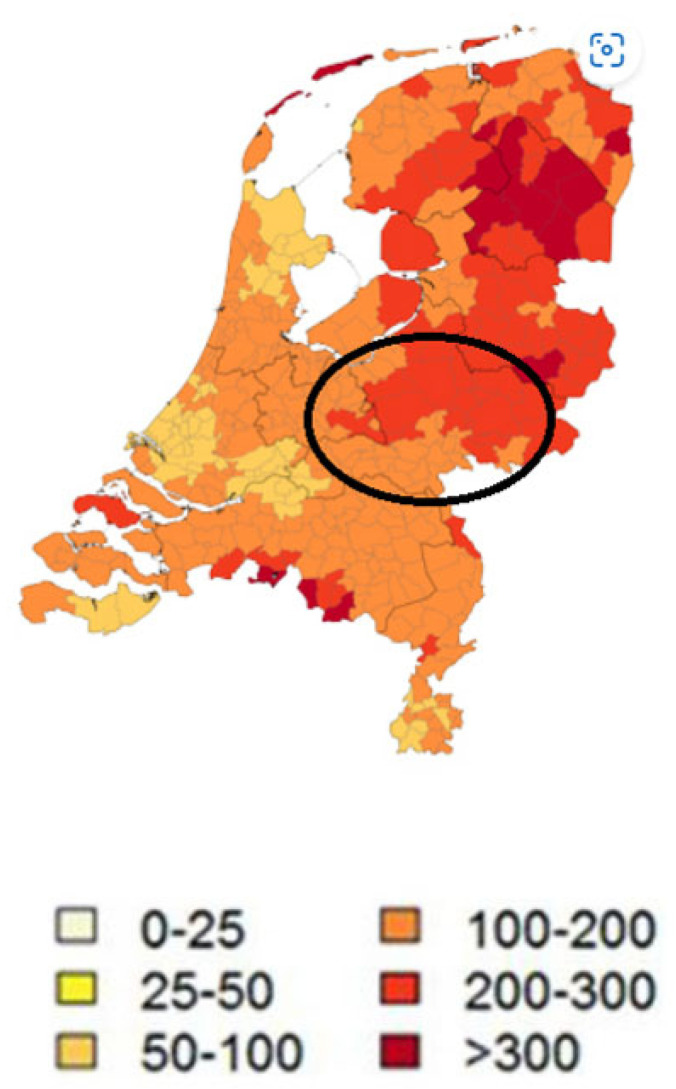
The region in the Netherlands where the forestry workers were employed is circled. Dark red regions are known for their higher incidence of Borrelia infection. The ranges refer to the number of diagnoses of erythema migrans per 100,000 inhabitants per year.

**Figure 3 life-13-01143-f003:**
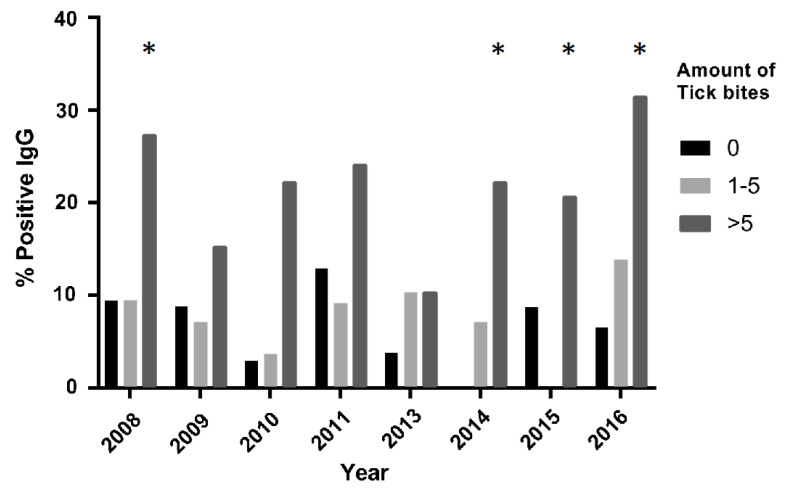
IgG seropositivity by year in relation to the number of tick bites (* = *p* < 0.05).

**Figure 4 life-13-01143-f004:**
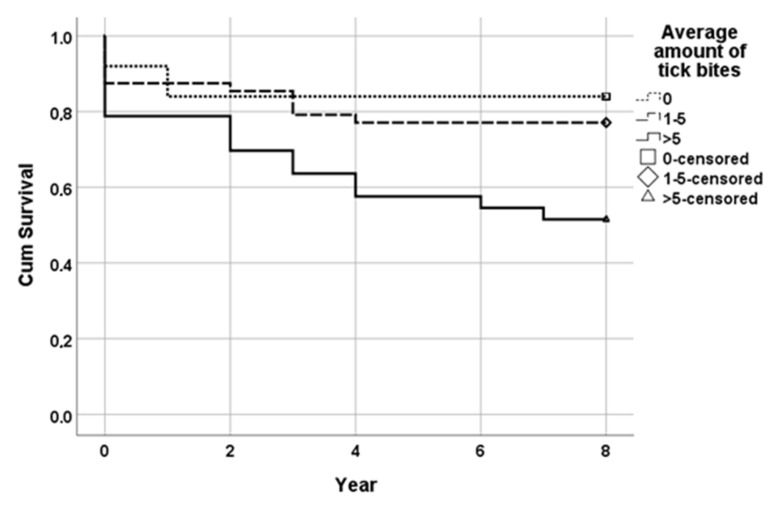
Cumulative survival model (Kaplan–Meier) of time to IgG seroconversion in relation to tick bites.

**Figure 5 life-13-01143-f005:**
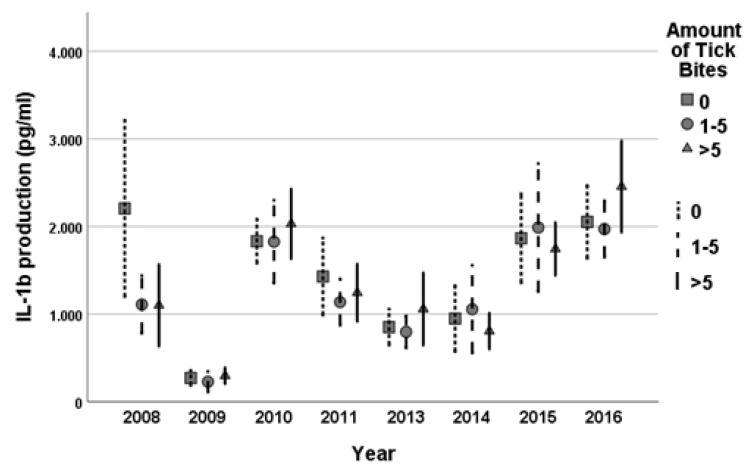
Clustered error bars (95% CI) of mean *Borrelia burgdorferi* 1 × 10^6^ pg/mL 24 h IL-1b PBMC by year and number of tick bites. 1.000 represents 1000 pg/mL.

**Table 1 life-13-01143-t001:** Population characteristics, seroprevalence, number of tick bites and forest exposure per year of follow-up.

Age in Years (Median Range)	46 (21–65)										
Gender, Male (%)	83.0										
Smoking, Yes (%)	24.0										
Type of Work (%):											
- Mainly inside	28.3										
- Mainly outside	46.2										
- Unknown/Unclear	25.5										
		Seroreactive			Number of tick bites				Forest Exposure (h/day)		
Year and Month of Sampling	Serology (number)	IgG (%)	IgM (%)	Missing (%)	0 (%)	1–5 (%)	>5 (%)	Missing (%)	<1.5 h (%)	1.5–5 h (%)	>5 h (%)
November 2008	98	14.2	27.3	7.5	30.2	31.1	32.1	6.6	17.9	20.8	59.4
November 2009	105	12.3	15.8	0.9	32.1	26.4	31.1	10.4	22.1	24.2	53.7
November 2010	105	12.3	35	0.9	33	26.4	25.5	15.1	20.5	26.1	53.4
November 2011	101	16	22.7	4.7	29.2	32.1	28.3	10.4	21.1	23.2	55.8
November 2012	–	–	–	–	–	–	–	–	–	–	–
November 2013	100	10.4	47.1	5.7	24.5	37.8	28.3	10.4	24.2	23.1	52.7
November 2014	99	13.2	15	6.6	31.1	26.4	17.9	24.5	24.4	20.5	55.1
November 2015	95	12.3	15.8	10.4	22.6	22.6	33	21.7	–	–	–
November 2016	97	17	25	8.5	30.2	27.4	34.9	7.5	26.9	20.5	52.6

**Table 2 life-13-01143-t002:** Characteristics of participants that seroconverted (n = 27) versus participants that did not seroconvert (n = 79).

	Not Seroconverted	Seroconverted	
Age (median range)	43 (21–65)	46 (23–64)	0.188 ^#^ p = 0.67
Gender, Male	79.5%	95.7%	3.3 * p = 0.07
Smoking	23.5%	26.1%	0.068 * p = 0.79
Number of Tick Bites			
0	30.1%	21.7%	
1–5	44.6%	13.0%	13.9 *
>5	25.3%	65.2%	p = 0.003

* Pearson’s chi-squared test; ^#^ *t*-test statistic.

## Data Availability

Data sharing not applicable due to privacy issues.

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
