# Peer review of "Lyme Borreliosis Serology: A Prospective Cohort Study of Forestry Service Workers in the Netherlands over 8 Years (2008 to 2016) of Follow-Up"

_life, 2023, doi:10.3390/life13051143_

Round 1

Reviewer 1 Report

The manuscript under review is an interesting study on the kinetics of anti-borrelia antibodies in forestry service workers. The authors wanted to analyze this in association to tick bites. The manuscript is interesting and well conducted, however I have a few comments:

-          The abstract should be rewritten: the background part needs to be implemented while the methods should be more synthetized. The background part as it is now only reports  the aim of the study, but does not provide any relevant information on what has been already studied or known about the topic.

-          In the introduction, first lines, the authors wrote “lyme disease is … most commonly caused by spirochaetal bacteria..”: is this the most common cause of lyme disease? Which are the other causes?

-          In materials and methods, first lines, “one hundred and six” should be written in numbers,

-          The conclusions should be implemented, as they are, they appear too short.

-          The conflicts of interest statement, funding, etc part should be adjusted according to the editor roles.

Author Response

Dear Reviewer,

Thank you for your comments. 

Please see attachment for our answers.

kind regards,

Hadewych ter Hofstede

Reviewer 2 Report

In the manuscript entitled "Lyme serology in the Netherlands: An 8-year follow up of forestry service workers", in which the authors have performed a retrospective serological analysis of blood samples from one-hundred-and-six forestry service workers for 8 years. The study is interesting. However, several issues need to be fixed before considering the manuscript for publication. 

1. Remove the kit details from the abstract section.

2. In several places for decimals, the comma (,) should be replaced with '.' .

3. What is the definition of seroconversion? should be explained in the introduction section with appropriate references.

4. What is the correlation between smoking vs seroconversion against Lyme disease? need to explain in detail.

5. Average amount of forest exposure time in hours should be explained.

6. The title should be modified appropriately to describe the type of study and study period.

7. Schematics for experimental design, number of participants in each year, the time of sample collection, antibody titers, etc should be given.

8. Individual antibody data for each of the 8- years should be given as a supplementary data file.

9. Representative western blotting figures for serum samples should be included. 

10. CMI assay method is incomplete. A reproducible method description should be given.

11. All tables and figures formatting should be improved.

12. The template of the questionnaire used can be included in the supplementary file section.

13. Legends for each table should be included.

14. What is the total number of participants in the study? Serology? Why there is a mismatch?

15. Is there any viral diseases that cross-react along with Lyme disease?

16. What is the significance of the survival model? Explain in detail.

17. Result of tick-repellent clothing - details not given in the methods and results section.

18. A geographical map showing the areas of sample collection from the forest worker in the Netherlands should be given.

19. The manuscript should be revised for the English language. 

Author Response

Dear Reviewer,

Thank you for your comments.

Please see the attachment for our answers.

kind regards,

Hadewych ter Hofstede

Round 2

Reviewer 1 Report

The manuscript has improved after the revisions. 

Author Response

No new changes have been made. 

Reviewer 2 Report

The authors have addressed most of the comments. Few minor changes are required. 

1. In the title and abstract, add the period of study - 2008 to 2016.

2. Add the scale bar for Figure 2. In terms of the number of cases or percentage of reported incidences. 

3.  Could not access or find the supplementary files. 

Author Response

  1. In the title and abstract, add the period of study - 2008 to 2016

Period 2008 to 2016 has been added.

2. Add the scale bar for Figure 2. In terms of the number of cases or percentage of reported incidences. 

See Figure: number of EM cases are added.

3.  Could not access or find the supplementary files. 

Files were previously added in ZIP file, now seperately added.